# FrontierCS: Evolving Challenges for Evolving Intelligence

**Qiuyang Mang** [* 1]   **Wenhao Chai** [* 2]   **Zhifei Li** [* 1]   **Huanzhi Mao** [* 1]   **Shang Zhou** [* 3]   **Alexander Du** [* 1 4]
**Hanchen Li** [* 1]   **Shu Liu** [* 1]   **Edwin Chen** [5]   **Yichuan Wang** [1]   **Xieting Chu** [6]   **Zerui Cheng** [2]   **Yuan Xu** [4]   **Tian Xia** [1]
**Zirui Wang** [1]   **Tianneng Shi** [1]   **Jianzhu Yao** [2]   **Yilong Zhao** [1]   **Qizheng Zhang** [7]   **Charlie F. Ruan** [1]   **Zeyu Shen** [2]
**Kaiyuan Liu** [8]   **Zhaoyang Hong** [9]   **Alex Gu** [10]   **Ziyi Zhang** [11]   **Runyuan He** [1]   **Dong Xing** [4]   **Zerui Li** [4]
**Zirong Zeng** [1]   **Yige Jiang** [12]   **Lufeng Cheng** [13]   **Ziyi Zhao** [9]   **Youran Sun** [1]   **Suyang Zhong** [14]   **Junpeng Wang** [15]
**Donglin Li** [5]   **Wenyuan Huang** [16]   **Jialiang Gu** [17]   **Wesley Kai Zheng** [1]   **Wangmeiyu Zhang** [5]   **Ruyi Ji** [18]
**Xuechang Tu** [6]   **Zihan Zheng** [19]   **Zhaozi Wang** [19]   **Zexing Chen** [3]   **Jingbang Chen** [§ 20]   **Jialu Zhang** [§ 21]
**Aleksandra Korolova** [§ 2]   **Peter Henderson** [§ 2]   **Pramod Viswanath** [§ 2]   **Vijay Ganesh** [§ 6]   **Saining Xie** [§ 19]
**Zhuang Liu** [§ 2]   **Dawn Song** [§ 1]   **Sewon Min** [§ 1]   **Ion Stoica** [§ 1]   **Joseph E. Gonzalez** [§ ¶ 1]   **Jingbo Shang** [§ ¶ 3]
**Alvin Cheung** [§ ¶ 1]

## Abstract

We introduce FrontierCS, a benchmark of 240 open-ended problems across diverse areas of computer science, designed and reviewed by experts, including CS PhDs and top-tier competitive programming participants and problem setters. Unlike existing benchmarks that focus on tasks with known optimal solutions, FrontierCS targets problems where *the optimal solution is unknown, but the quality of a solution can be objectively evaluated.* Models solve these tasks by implementing executable programs rather than outputting a direct answer. FrontierCS includes algorithmic problems, which are often NP-hard variants of competitive programming problems with objective partial scoring, and research problems with the same property. For each problem, we provide an expert reference solution and an automatic evaluator. Combining open-ended design, measurable progress, and expert curation, FrontierCS provides a benchmark at the frontier of computer-science difficulty. Empirically, we find that frontier reasoning models still lag far behind

human experts, and that simply increasing reasoning budgets does not close this gap on open-ended challenges. Moreover, these models struggle to identify internal equivalence classes, and existing agentic frameworks exhibit brittleness on such problems due to overfitting. FrontierCS thus offers a new lens into model capabilities on real frontier computer science problems.

## 1. Introduction

The rapid progress of large language models (LLMs) is evident on numerous code and reasoning benchmarks (Mao et al., 2025; Patil et al., 2025; Jain et al., 2024; Zheng et al., 2025; Ouyang et al., 2025; Xu et al., 2025; Wang et al., 2025b; Quan et al., 2025; Chen et al., 2021; Zhuo et al., 2025; Liu et al., 2023b; Ding et al., 2023; Liu et al., 2023a; Li et al., 2022; Xia et al., 2025; Yang et al., 2025; Fang et al., 2024), which largely comprise closed-form tasks with a single optimal answer and a pass-or-fail criterion. Yet many frontier problems in computer science are intrinsically open-ended, requiring nuanced trade-offs among quality, efficiency, and robustness (Chen et al., 2025b; Nie et al., 2025; Fan et al., 2024; Chen et al., 2025a; Li et al., 2025a; Ma et al., 2025a). While recent work has begun using LLMs to address unsolved or open-ended CS problems (Cheng et al., 2025; Ma et al., 2025b; Novikov et al., 2025; Agrawal et al., 2025; Mang et al., 2025), these efforts naturally evaluate models only within their specific application domains or on a small number of representative cases, leaving the field without a comprehensive, cross-domain benchmark.

In this paper, we introduce FrontierCS, a coding benchmark that evaluates LLMs on solving open-ended computer science problems, where no known closed-form or deterministic optimal solution exists in practice. Unlike math or reason-

---

[*]Equal contribution. [§]Advising. [¶]Equal advising. [1]UC Berkeley [2]Princeton University [3]UCSD [4]X-camp Academy [5]Independent [6]Georgia Tech [7]Stanford University [8]University of Washington [9]UIUC [10]MIT [11]University of Chicago [12]Nanyang Technological University [13]University of Toronto [14]National University of Singapore [15]Duke University [16]University of Wisconsin-Madison [17]George Mason University [18]University of Michigan [19]New York University [20]Universal Cup [21]University of Waterloo. Correspondence to: Qiuyang Mang <qmang@berkeley.edu>, Wenhao Chai <wenhao.chai@princeton.edu>.

*Proceedings of the $43^{rd}$ International Conference on Machine Learning*, Seoul, South Korea. PMLR 306, 2026. Copyright 2026 by the author(s).

ing benchmarks that require a direct answer (e.g., AIME), FrontierCS requires models to implement executable programs to solve the problem (e.g., LiveCodeBench). We focus on tasks where a global optimum is either unknown or practically unattainable, yet any proposed solution can be deterministically checked for validity and assigned a score by an automatic evaluator (Li et al., 2025b; Imajuku et al., 2025; Wang et al., 2025c). This design measures the ability of models to implement effective and efficient algorithms rather than algorithms that exhaustively solve the problem. FrontierCS includes optimization tasks for which no known optimal solver exists in practical time, each task includes an expert reference solution and a deterministic automatic evaluator to facilitate objective comparisons and ensure reproducibility. Each FrontierCS task can be solved by submitting executable code: the evaluator runs the program on generated instances and scores its outputs by task-specific metrics under resource limits (*e.g.,* time and memory usage). The model is prompted with the problem specification (and any required I/O or API stubs) and must produce a self-contained solver program.

As an example of the type of tasks included in FrontierCS, consider the Polyomino Packing task in Figure 1, packing a set of $n$ polyominos (block shapes) into the smallest possible grid without overlapping to maximize density, *i.e.,* the area fraction occupied by the polyominoes. Even though the optimal packing is unknown, solutions can still be compared objectively using packing density. The evaluator first runs the LLM-generated program to obtain a packing arrangement, then checks whether the arrangement is valid, *i.e.,* if any polyominoes overlap or extend beyond the grid boundaries. If valid, the score is computed as the packing density, defined as the total area covered by all placed polyominoes divided by the area of the grid. For this task, both the human expert and GPT-5 produce valid packings, but the human achieves 87% density while GPT-5 achieves only 47%. By varying $n$, this single specification yields infinitely many task instances, each with its own difficulty and unknown optimum. This example illustrates the kind of tasks targeted by FrontierCS: problems with many valid solutions whose quality spans a continuous spectrum rather than a simple pass-or-fail outcome.

Formally, we define an open-ended optimization problem without known polynomial time solutions as one that satisfies the following:

- **Unsolved or Intractable Optimum:** The global optimum is unknown to compute over all problem instances, requiring progress to come from creative algorithms, heuristics, search, or optimization.

- **Deterministically Verifiable and Quantitatively Scored:** Solutions with runtime limit can be automatically checked for validity and assigned a numeric score that reflects the

quality of the solution, rather than a simple pass-or-fail.

- **Parametric Problem Generator:** The task specification induces a large, variable-difficulty space of instances, enabling unseen test cases to prevent leak and overfitting.

FrontierCS focuses on two tracks: algorithmic optimization problems, and tasks more closely tied to real-world CS research. Both tracks naturally exhibit open-endedness, stress-testing a model's ability to perform deep open-ended reasoning and discover nontrivial optimization strategies. The design of FrontierCS encourages iterative improvement in an open-ended landscape rather than aiming for a deterministic optimal solution, since none of its problems have known practical optima. FrontierCS, with its dynamic task scaling and objective quantitative feedback, offers an adaptive framework for continuous progress in LLM reasoning and creativity. Moreover, given that solutions are automatically verifiable and reward signals are available, FrontierCS is well-suited not only for evaluation but also for training and ablation studies. The scoring functions can effectively drive reinforcement learning (Zhou et al., 2025; Madaan et al., 2023; Chen et al., 2025c; Fang et al., 2025; Liu et al., 2025; van Niekerk et al., 2025; Zweiger et al., 2025; Zhao et al., 2025; Wang et al., 2023) as reward model.

Empirically, we find that even the strongest frontier reasoning models remain far behind human experts on both the algorithmic and research tracks of FrontierCS. Simply scaling up context length or reasoning budgets yields diminishing returns on the hardest problems, and models frequently converge to locally workable but clearly suboptimal algorithms. These observations suggest that current large reasoning models are still missing key ingredients for truly open-ended computer-science problem solving, motivating FrontierCS as a challenging benchmark for future progress.

## 2. Related Work

**Closed-form benchmarks for code and reasoning.** A wealth of benchmarks have evaluated LLMs on coding and math problems with known solutions. In the programming domain, tasks like HumanEval (Chen et al., 2021), MBPP (Austin et al., 2021), SWE-bench (Jimenez et al., 2023), BFCL (Patil et al., 2025), and LiveCodeBench (Jain et al., 2024) present coding challenges with unit tests as a pass-or-fail criterion. These were useful for early code models, but top LLMs now approach saturation on them. Similarly, in math and logical reasoning, datasets such as MATH (Lightman et al., 2023), GSM8K (Cobbe et al., 2021), and AIME were designed to test step-by-step reasoning. State-of-the-art models have achieved near-perfect scores on these high-school-level benchmarks, indicating that they no longer discriminate well at the frontier of ability. In the domain of mathematics, FrontierMath (Glazer

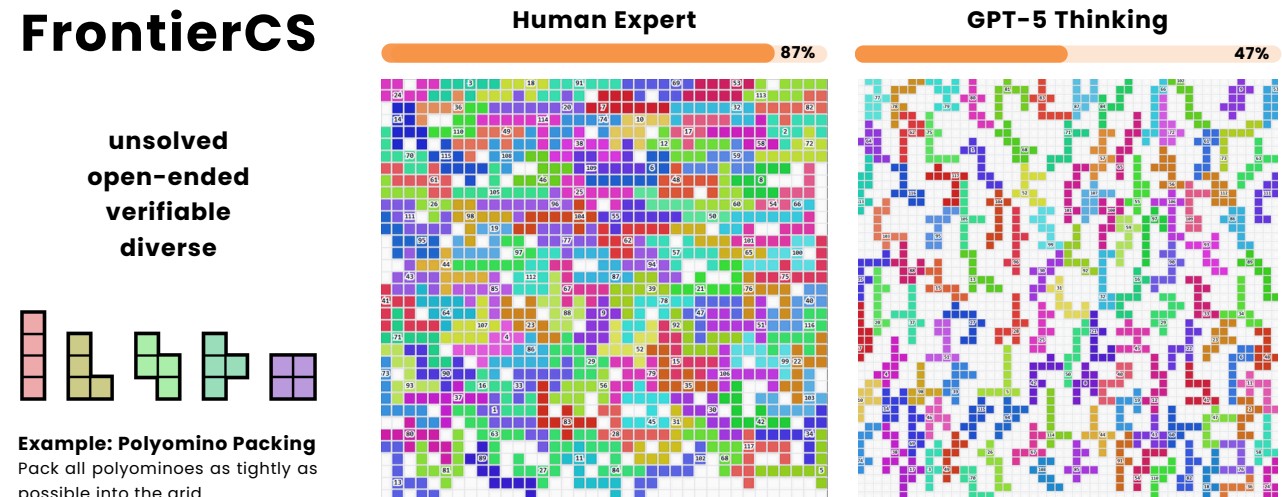

*Figure 1.* **FrontierCS**, an unsolved, open-ended, verifiable, and diverse benchmark for computer science tasks. The Polyomino Packing example, where both human experts and LLMs produce valid but non-optimal solutions that differ substantially in density. This reflects the benchmark's core design choice: problems are unsolved, admit many solution strategies, and are evaluated via deterministic scoring rather than pass-or-fail.

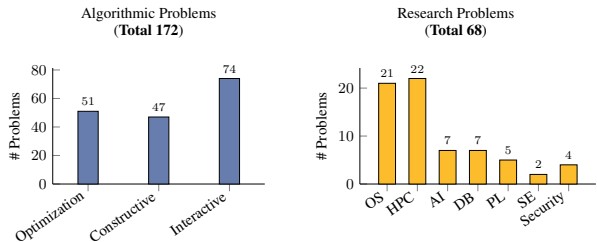

*Figure 2.* **Categories** distribution of the 240 problems in FrontierCS. **Left**: Algorithmic Problems, covering Optimization tasks, Constructive tasks, and Interactive tasks, adapted from programming contests but rewritten into open-ended, partially scored variants. **Right**: Research Problems, spanning six major CS domains: OS (Operating Systems), HPC (High-Performance Computing), AI (Artificial Intelligence research tasks), DB (Databases), PL (Programming Languages), and Security (cybersecurity and vulnerability analysis). These research problems are sourced from real research workflows.

et al., 2024) collects hundreds of new, expert-crafted math challenges spanning areas from algebraic geometry to number theory, each designed to require substantial creativity and insight. Crucially, every problem has a reference answer and an automated solution checker, allowing objective scoring even if the solution process is complex. Recently, LiveCodeBench Pro (Zheng et al., 2025) has increased the difficulty of its competitive programming benchmark and updates it quarterly. On the hardest split, only a few models are able to solve even a single problem. Even so, LiveCodeBench Pro still focuses only on problems that have known optimal solutions, whereas in many real-world scenarios, finding the optimal solution is tied to NP-completeness.

**Open-ended and partially-scored benchmarks.** Beyond strictly one-answer tasks, a number of benchmarks embrace open-ended problem solving or partial-score evaluation. ALE-Bench (Imajuku et al., 2025) introduces optimization-style tasks from AtCoder Heuristic Contests, replacing binary correctness with score-based evaluation. Spanning routing, packing, scheduling, and stochastic search, it challenges models to design heuristics that achieve high objective scores under strict runtime constraints. UQ (Nie et al., 2025) curates a set of unsolved tasks from a collection of over 500 problems on forums like Stack Exchange, showing that even top models pass only around 15% after human validation. MLR-Bench (Chen et al., 2025b) evaluates models on research tasks taken from top ML conferences, using an automated judge and agent. MLR-Bench finds that models are capable of writing coherent papers but still struggle with experimentation. Additionally, their automated judge aligns closely with human reviewers, revealing the possibility of automated evaluation for research tasks. NP-Engine (Li et al., 2025b) introduces a benchmark and training framework for 10 classic NP-hard tasks, including Subset Sum, the Traveling Salesman Problem, and others. A model trained on their framework achieves state-of-the-art performance on these tasks and shows strong out-of-domain generalization. KernelBench (Ouyang et al., 2025) evaluates LLMs on writing efficient GPU kernels for a variety of computational tasks, using performance metrics like correctness and runtime.

Existing benchmarks generally adopt one of three evaluation paradigms: binary pass-or-fail testing (*e.g.,* LiveCodeBench), performance- or runtime-based scoring (*e.g.,* KernelBench), or open-ended tasks that require human or

LLM-as-judge evaluation (*e.g.,* UQ). Although ALE-Bench and the AtCoder Heuristic Contests fall outside these categories by using optimization-style scoring, they still cover a narrow slice of problem types, and their tasks are authored by only a small number of recurring problem setters, limiting their diversity. FrontierCS addresses these gaps by providing a large, diverse, and continually expanding collection of real-world reasoning problems sourced from multiple platforms. Each problem is unsolved yet automatically verifiable, enabling objective evaluation without relying on binary correctness or human judgment. Instead of checking only for optimality or execution performance, FrontierCS supports continuous scoring based on solution quality, allowing models to be credited for partial progress on research-style tasks where optimal solutions are unknown or computationally prohibitive. Together, these properties establish FrontierCS as a new baseline for evaluating frontier model reasoning in open-ended, real-world problem settings.

Recent works like ThetaEvolve (Wang et al., 2025a) and TTT-Discover allows models to continually learn from their experiences in improving open optimization problems. However, these efforts are typically demonstrated on a small number of handpicked optimization problems, whereas FrontierCS provides a large and diverse suite of automatically verifiable optimization tasks to support systematic evaluation and scaling.

## 3. Problem Collection

As shown in Figure 2, FrontierCS consists 240 problems across two tracks: Algorithmic Problems and Research Problems. The Algorithmic Problems track contains 172 problems adapted from programming contests, covering three categories: Optimization, Constructive, and Interactive problems. The Research Problems track contains 68 problems sourced from real-world computer science research questions, spanning 7 domains. Each problem is carefully curated through a multi-stage process involving proposal, implementation, and review to ensure quality and relevance, which will be detailed in the following sections.

**Taxonomy.** To organize the algorithmic problems in FrontierCS, we adopt a taxonomy that reflects the dominant reasoning mode each problem requires. **(1)** *Constructive problems* center on synthesizing a valid structured object (*e.g.,* graph and packing) under global constraints; even when tasks ask for a minimal or smallest solution, the difficulty lies in producing a coherent structure rather than tuning explicit parameters (*e.g.,* Problem 1, Problem 4 and Problem 5 in Section A). **(2)** *Optimization problems*, in contrast, require explicit search over a parameterized space to minimize or maximize a quantitative objective, often involving accuracy-latency or cost-capacity trade-offs (*e.g.,* Problem 2 in Section A). **(3)** *Interactive problems* involve

solving a hidden-instance task through a query-response protocol, where each action depends on previous replies, and performance typically depends not only on correctness but also on interaction efficiency such as minimizing the number of steps (*e.g.,* Problem 3 and Problem 10 in Section A). Together, these three categories capture the principal forms of algorithmic and open-ended reasoning that arise in real computer science tasks.

For problems in the Research Problems track, we categorize them based on their respective computer science domains, including Operating Systems (OS), High-Performance Computing (HPC), Artificial Intelligence (AI), Databases (DB), Programming Languages (PL), Software Engineering (SE), and Security. Specifically, when a research topic spans multiple domains, we assign it to the domain that the model's solution is primarily meant to address. This domain-based taxonomy reflects the diverse challenges and methodologies inherent in different areas of computer science research.

### 3.1. Algorithmic Problems

Our algorithmic problems largely originate from programming contests (*e.g.,* Codeforces, AtCoder, ICPC, IOI) and classical CS problem settings (*e.g.,* knapsack), where tasks are solved under time and memory limits and are automatically judged. However, most contest problems admit a single optimal solution and are scored with a binary pass-or-fail system, which does not meet our requirement for open-ended problems without a known optimum. To address this mismatch, we introduce a structured curation pipeline, *i.e.*, Proposal, Implementation, and Review, to construct genuinely variant.

**Proposal.** This stage is led by experts with qualifications equivalent to ICPC World Finalists, who are responsible for submitting candidate problems, complete with links to their original sources and a description of intended modifications. The proposal is reviewed by experts against: (i) openness and multiplicity of viable solutions; (ii) discriminative strength of the scoring scheme; (iii) clarity and completeness of the problem statement and data ranges.

**Implementation.** This stage: (i) converts each problem into an open-ended variant by changing single-optimum objectives and a partial scoring system; (ii) standardizes input/output formats or, when appropriate, provides an interaction library; (iii) implements a deterministic verifier to formally assess the validity of candidate solutions; and (iv) delivers a SOTA human reference solution; (v) provides the necessary configuration and test data for evaluation.

**Review.** After implementation, each problem is reviewed by another algorithmic-problem expert to make sure that: (1) the problem has no known optimal solution or a nearly optimal solution that leaves limited room for improvement; (2) the scoring policy is objective and can meaningfully reflect progress; (3) the human reference solution is signifi-

cantly stronger than the best model's performance; and (4) the evaluator is implemented correctly, and the test data can comprehensively evaluate solutions.

In total, FrontierCS contains 172 high-quality algorithmic problems adapted from programming contests, including 51 optimization problems, 47 constructive problems, and 74 interactive problems. Each problem contains: a well-defined statement with formal constraints; generators for test cases; a deterministic verifier with a partial scoring system; an expert-authored reference solution; a suite of baseline implementations; a reproducible model evaluation harness with associated scripts and metrics; and thorough documentation on provenance and data decontamination.

**Scoring policy design.** We primarily evaluate solutions using problem-specific quality metrics such as cost, density, or number of queries, rather than computational efficiency. This reflects the open-ended nature of our tasks, where the main challenge is designing effective algorithms or strategies. Like traditional competitive programming tasks, each problem includes strict time and memory limits, and any solution that exceeds these limits is considered invalid and receives no score. In this way, runtime and memory act as feasibility constraints rather than components of the scoring metric. Unless explicitly specified otherwise, runtime is never part of the scoring, and together with the resource limits, this ensures that higher scores reflect better strategies rather than solutions that rely on excessive compute.

**On contest subtasks vs. our scoring.** In programming contests such as ICPC and IOI, *subtasks* partition tests into groups with extra constraints; points are awarded only if *all* cases in a subtask pass, yielding discrete, all-or-nothing partial credit. Moreover, these subtasks are typically not strongly related to the essence of the problem, such as simplifying the problem's constraints or addressing the input data size. Our benchmark instead uses task-specific, objective metrics (*e.g.,* density, cost, number of queries) with continuous or piecewise-continuous mappings to scores, measured relative to a trivial baseline and a strong human reference. This design rewards incremental improvements and enables fair comparison even when no single optimal solution is known.

### 3.2. Research Problems

**Proposal.** For FrontierCS research problems, we ask CS PhD students to set problems based on their unsolved research questions and to implement the evaluation container. Some of the research problems we select are drawn from recent AI-Driven Research for Systems (ADRS) work (Cheng et al., 2025), including multi-region spot-instance scheduling, cloud transfer path optimization, and optimizing SQL queries with LLM calls. Similarly to the algorithmic problems, we use a process of *proposal* and *implementation*,

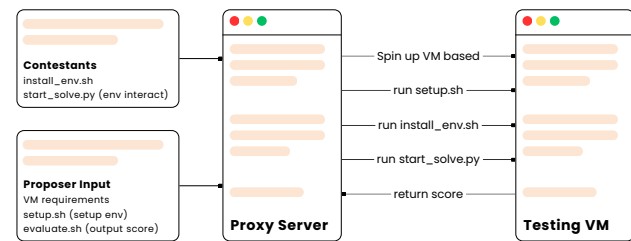

*Figure 3.* **Evaluation pipeline** of FrontierCS research problem using SkyPilot (Yang et al., 2023)

with *review* stages in between.

**Implementation.** Unlike problems sourced from programming contests that can be judged in a unified framework, each research problem needs a specialized environment. This stage ensures full reproducibility and objective evaluation. Candidate research problems contains resources, set_up_env.sh, evaluate.sh, and a README file.

The README describes the problem statement, VM or Docker requirements, the data layout prepared by environment setup script, and the exact input/output contract of the participant's solution. The environment can be utilized by the LLM or agent to create its solution for the problem. Examples include a training set for image classification problems. Last but not least, the problem must support automated scoring via `evaluate.sh`, which gives a standardized score between 0 and 100. The evaluation has to be deterministic and must not use an LLM as a judge. An overview of the architecture of the research evaluation platform is shown in Figure 3.

**Infrastructure.** To facilitate extensive experimentation, we integrate SkyPilot (Yang et al., 2023) to manage the compute infrastructure. This abstraction allows the evaluation to scale up seamlessly from a single node to distributed cloud clusters, handling the heterogeneous hardware requirements of our research track. Crucially, it optimizes for economic viability by leveraging spot instances and region arbitrage, enabling researchers to evaluate agents on the full benchmark at minimal costs.

**Review.** Each submission is reviewed by CS researchers according to the following criteria: (i) the problem should not have a single optimal solution; (ii) partial scoring system should meaningfully reflect progress; (iii) all scripts must run unattended in a fresh VM; (iv) the environment must be deterministic, isolated, and free from external dependencies.

**Scoring policy design.** Unlike algorithmic problems, which are scored primarily on solution quality, research problems often involve multiple objectives such as accuracy, latency, memory usage, and cost due to their real-world nature. For example, a task may require designing a vector

database index that minimizes query latency while maintaining at least 95% accuracy (Problem 7 in Section A). Nevertheless, we still impose strict resource limits for each problem to prevent excessive computation, and any solution exceeding the limit is considered invalid with no score.

**Variants.** For each research problem, we provide multiple variants with different resource constraints and objective targets to reflect real-world research scenarios. For example, a variant may impose a stricter memory limit but lower the accuracy requirement, changing the hardware setting from CPU to GPU, or adjusting the objective from latency minimization to throughput maximization. Note that, we treat every variant as an independent problem in the reported totals, because differing resource constraints or objectives lead to distinct solution strategies and reflect how related tasks are separately evaluated in real computer science research.

In total, FrontierCS contains 68 research problems across diverse domains such as symbolic regression, vector database design, and kernel optimization. Each accepted research problem includes: (i) a detailed problem description and background motivation; (ii) a reproducible environment with pinned dependencies or a Docker image; (iii) a deterministic evaluator with partial scoring and diagnostics; (iv) an expert-authored reference solution; (v) the most trivial baseline implementations for comparison.

### 3.3. Update Policy

A core design principle of FrontierCS is to enable measurable progress on open-ended tasks. Rather than using binary judgments, our evaluators provide a score of $0 - 100$ using human performance as the baseline, allowing for measurable feedback on incremental improvements. FrontierCS is designed to remain relevant as models improve by supporting three complementary forms of task evolution.

**(1) Adding new tasks.** When expanding the scope of the benchmark or introducing fundamentally new problem categories, we may add new tasks. This is the traditional route for extending a benchmark, but it is not the only mechanism FrontierCS provides.

**(2) Increasing the difficulty of existing tasks without changing the problem statement.** A key feature of FrontierCS is the separation between the problem statement and the metric goal. This decoupling allows us to preserve the original question text while making the task more challenging. Difficulty can be increased by tightening constraints (e.g., time or memory budgets, feasibility requirements), modifying workloads or datasets (e.g., larger or more adversarial instances), or adjusting optimization objectives (e.g., stricter accuracy or performance targets). These updates retain task continuity while ensuring that the benchmark keeps pace with advancing model capabilities.

**(3) Refining human reference solutions and evaluation thresholds.** When models approach or surpass strong human baselines, we can refine the human reference solution, scoring rubric, or evaluation thresholds to provide finergrained separation between capable models. This method increases difficulty without modifying either the task description or the environment.

## 4. Evaluation Results

In this section, we report the metric design and the performance of the frontier models on FrontierCS.

### 4.1. Setup

In our evaluation, we tested 7 frontier models: GPT 5 Thinking (OpenAI, 2025), GPT 5.1 Thinking, GPT 5.2 Thinking, Gemini 2.5 Pro (DeepMind, 2025a), Gemini 3.0 Pro (DeepMind, 2025b), Grok 4 (xAI, 2025), and DeepSeek V3.2 Thinking (DeepSeek-AI, 2025). We compared model performance against human experts on both the algorithmic and research problem tracks. For each LLM request, we enforced a time-out of 20 minutes. For GPT-5, GPT-5.1, GPT-5.2 (Thinking), and Grok 4, we set reasoning_effort to high. For Gemini 2.5 Pro and Gemini 3.0 Pro, we set thinking_budget to -1 (default).

During evaluation, models are tested in a single-round setting: once they produce a solution, that output is final. They do not have the opportunity to run their code, inspect unittest results, or iterate based on feedback. They also do not have access to a code editor, a Python environment, or any external tools. All inputs are provided purely as text; *i.e.,* no diagrams or visual components are involved, and each problem can be fully described in text alone.

### 4.2. Metrics

In FrontierCS, no problem has a universally correct solution, so we cannot use accuracy as an evaluation metric. Instead, we introduce a grading system that scores each solution relative to two reference points: the reference solution written by human experts and a trivial baseline solution. Meanwhile, for some problems that have nontrivial bounds on achievable performance, we also use those bounds for reference. A detailed example can be found in Section A.

Specifically, for a test case, if a solution fails to reach the level of the trivial baseline, it receives a score of zero; if it surpasses the threshold defined by a human expert reference solution or the nontrivial bound, it receives a full score. Scores between these two extremes are assigned based on the nature of the problem, ensuring that each task has its own fair and tailored grading system.

We report both Score@1, Avg@5, and Score@5, as we

| Model | Score@1 | Avg@5 | Score@5 | Elo |
|---|---|---|---|---|
| Human Experts | **86.99** | - | - | - |
| Gemini 3.0 Pro | **33.12** | **34.58** | **56.09** | **1265** |
| GPT 5.2 Thinking | 32.40 | 33.11 | 47.19 | 1242 |
| GPT 5 Thinking | 23.10 | 22.58 | 39.73 | 1196 |
| DeepSeek 3.2 | 24.83 | 23.89 | 41.44 | 1193 |
| Grok 4 | 24.04 | 22.98 | 36.81 | 1174 |
| Gemini 2.5 Pro | 20.34 | 19.32 | 36.65 | 1167 |
| GPT 5.1 Thinking | 20.64 | 21.49 | 34.76 | 1164 |

*(a)* Algorithmic problems

| Model | Score@1 | Avg@5 | Score@5 | Elo |
|---|---|---|---|---|
| Gemini 3.0 Pro | **46.55** | **43.14** | **59.22** | **1283** |
| GPT 5 Thinking | 30.91 | 34.94 | 55.25 | 1218 |
| GPT 5.1 Thinking | 32.12 | 33.70 | 56.79 | 1214 |
| GPT 5.2 Thinking | 30.29 | 34.09 | 58.90 | 1210 |
| Gemini 2.5 Pro | 21.66 | 25.74 | 51.57 | 1180 |
| Grok 4 | 26.75 | 24.01 | 48.15 | 1149 |
| DeepSeek 3.2 | 21.51 | 21.76 | 44.41 | 1146 |

*(b)* Research problems

*Table 1.* **FrontierCS benchmark results.** We define Score@k as the highest score achieved across $k$ runs, Avg@k as the average score over those $k$ runs, and Elo ratings are computed using the Bradley-Terry model based on single-attempt performance, measuring relative model capabilities while normalizing for problem difficulty.

observe that the model exhibits stochasticity and notable improvement across multiple attempts. Here, we define Score@k as the maximum score among the $k$ model trials, and Avg@k as the average score among those trials.

**Elo Rating** We compute Elo ratings using the Bradley-Terry model based on single-attempt performance. For each problem, we first calculate the expected win rate between two models by averaging outcomes across all pairwise combinations of their submitted solutions. We adopt this pairwise approach to normalize differences in problem difficulty, ensuring the evaluation captures relative model capability rather than raw score magnitudes. These problem-level statistics are then aggregated via Maximum A Posteriori optimization with a Gaussian prior to estimate global ratings.

### 4.3. Quantitative Results

As shown in Table 1a, frontier models still lag significantly behind human experts on the algorithmic problems track, where the 7 frontier models achieve Score@1 of Gemini 3.0 Pro (33.12), GPT 5.2 Thinking (32.40), DeepSeek 3.2 (24.83), GPT 5 Thinking (23.10), GPT 5.1 Thinking (20.64), Grok 4 (24.04), Gemini 2.5 Pro (20.34), respectively, compared to human experts' 86.99. These results highlight the substantial gap that remains between current AI capabilities and human expertise in tackling open-ended algorithmic challenges. As expected, we also observe that all seven models demonstrate stronger performance when increasing the number of sampling attempts from one to five, indicating that they can generate diverse solutions and benefit from multiple trials. For instance, Score@5 improves over Score@1 by 12.77 – 22.97 points.

For the research track, Table 1b reports the same metrics as the algorithmic track across 7 frontier models. Gemini 3.0 pro attains the best results on all metrics. Across models, additional sampling also yields sizable gains, *i.e.,* Score@5 improves over Score@1 by 12.67 – 29.91 points.

## 5. Discussion

In this section, we conduct additional experiments and analyses on FrontierCS to better understand model behaviors and failure patterns, and to extend our evaluation to an agentic framework, *i.e.,* OpenEvolve (Sharma, 2025).

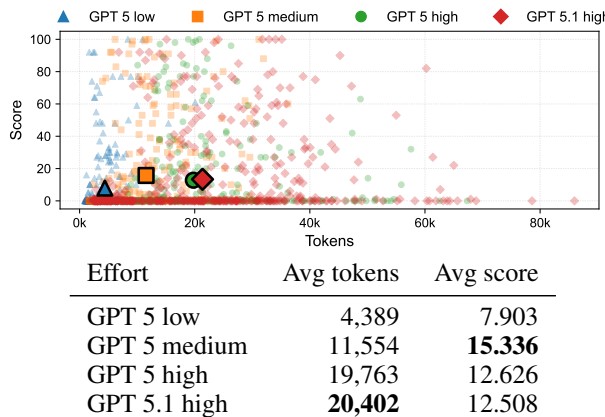

| Effort | Avg tokens | Avg score |
|---|---|---|
| GPT 5 low | 4,389 | 7.903 |
| GPT 5 medium | 11,554 | **15.336** |
| GPT 5 high | 19,763 | 12.626 |
| GPT 5.1 high | **20,402** | 12.508 |

*Figure 4.* **Reasoning tokens vs. Score**: We are setting GPT 5 Thinking with different reasoning efforts in a subset. **Top**: scatter plot; Each point is one attempt (3 attempts per problem). Marks in the middle denote group averages. **Bottom**: average tokens and scores by reasoning effort.

### 5.1. Improving Reasoning Effort Fails

We further analyze the relationship between reasoning effort and model performance on open-ended algorithmic problems. As shown in Figure 4, we plot GPT 5 Thinking's score against the number of reasoning tokens it consumes per attempt by setting different `reasoning_effort` levels, *i.e.,* `low`, `medium`, and `high` in a subset. As expected, we observe a clear positive correlation between reasoning effort when comparing low and medium reasoning levels. However, increasing the reasoning effort from medium to high does not yield further gains; in fact, performance drops from 15.336 to 12.626, suggesting diminishing returns at higher reasoning budgets. This indicates that while increased rea-

soning effort generally aids performance on open-ended problems, there may be an upper limit beyond which additional effort yields limited benefits for current LLMs. Future work could explore more effective ways to leverage high reasoning effort for complex open-ended problem solving.

### 5.2. Misleading Micro-Optimization Trap

During evaluation, we identify a recurring failure pattern in LLM behavior: the model often fixates on small, low-impact optimizations while overlooking the core algorithmic choices required for substantial performance gains. This is especially evident in Polyomino Packing (Problem 5 in Section A), which asks models to pack numerous polyominoes into a minimum-area rectangle and output a list of transformations for each piece. In practice, GPT 5-Thinking frequently adopts this transformation list as its internal data structure. Although memory-efficient and aligned with the final output format, this choice is a conceptual pitfall. Relying solely on transformation lists renders overlap detection and free-space search both cumbersome and error-prone. Consequently, the model produces invalid code in about 30% of attempts, and in the remaining 70% achieves only low scores, *i.e.,* 20 – 50 due to ineffective search strategies.

In contrast, a minimal prompt adjustment dramatically changes the outcome. Adding a single instruction like *Please use a 2D array to maintain the rectangle state, and convert to the required format only at the end* reliably shifts the model toward a structurally sound internal representation. With this modification, the zero-score rate drops to about 10%, and in nearly 80% of cases the model successfully implements an efficient search strategy, achieving high scores in the 80 – 85 range, consistently surpassing prior best solutions. This case study highlights a fundamental limitation of current LLMs: they do not inherently recognize which optimizations are algorithmically meaningful, often becoming trapped in superficially appealing but strategically irrelevant micro-optimizations.

### 5.3. Formulation Sensitivity in Open-ended Problems

Since many open-ended problems belong to the same equivalence classes (*e.g.,* Problem 11 and Problem 12), it is natural to investigate whether models can normalize different formulations into a shared problem class via reductions or whether they instead remain highly sensitive to surface-level problem formulations. In FrontierCS, we deliberately construct nine pairs of problems where one problem can be reduced to the other. To ensure a fair comparison, each test case in one problem is mapped via reduction to a semantically equivalent test case in its paired problem, and both are evaluated under the same scoring policy. The models' performance on those problems is shown in Table 2. As a result, all frontier models exhibit inconsistent performance

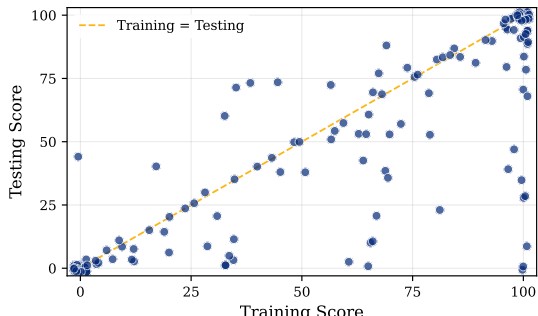

*Figure 5.* Training vs. Testing Performance of OpenEvolve on FrontierCS algorithmic track with GPT 5 Thinking. Points in the **bottom-right** region indicate training performance is higher than testing performance. The average training score is **50.75**, while the average testing score drops to **43.00**. Moreover, for **46%** of problems, the best program produced by OpenEvolve fails to outperform one-shot GPT-5 Thinking.

*Table 2.* Model performance on better vs. worse formulations of equivalent open-ended problems. For each model, the better and worse formulation within each equivalent problem pair is determined independently according to the model's score.

| Model | Better Set | | Worse Set | |
| --- | --- | --- | --- | --- |
| | Avg Score | Avg Std | Avg Score | Avg Std |
| DeepSeek 3.2 | 93.05 | 9.23 | 82.92 | 20.48 |
| GPT 5.1 Thinking | 88.02 | 7.07 | 78.76 | 8.49 |
| GPT 5.2 Thinking | 87.17 | 13.45 | 65.46 | 22.87 |
| GPT 5 Thinking | 86.29 | 11.81 | 70.58 | 28.68 |
| Grok 4 | 85.69 | 6.95 | 75.65 | 12.24 |
| Gemini 3.0 Pro | 83.79 | 12.64 | 67.21 | 20.15 |
| Gemini 2.5 Pro | 80.58 | 11.81 | 61.85 | 22.34 |

on open-ended problems that share the same underlying problem class but differ in formulation. Moreover, lower-scoring formulations are associated with higher variance in model performance. Overall, these results suggest that current frontier models remain highly sensitive to formulation for open-ended problems.

### 5.4. Overfitting in Agentic Evolving

We use OpenEvolve (Sharma, 2025) as a case study to evaluate the performance of an agentic framework on our open-ended problems. Specifically, we conduct experiments on 172 algorithmic problems using GPT 5 Thinking as the base model with a temperature of 0.7. For each problem, we randomly sample 30% of the test cases as the training set for evolution, and evaluate the best-evolved programs on the full test suite used in Section 4. As shown in Figure 5, OpenEvolve does not yield significant performance improvements when trained on only a 30% subset of test cases, and instead suffers from overfitting.

## 6. Limitation and Future Works

Our evaluation primarily measures one-shot coding performance: models are tested in a single-round setting without access to execution feedback, debugging, or iterative refinement. Future works will incorporate execution feedback would allow benchmarking agentic pipelines.

## 7. Conclusion

We introduce FrontierCS, a comprehensive and diverse benchmark for open-ended computer science tasks where global optima are unknown but solutions remain deterministically verifiable and partially gradable. FrontierCS provides expert reference solutions, an automated evaluator, and a fully reproducible pipeline, and adopts versioned difficulty scheduling to preserve discrimination as models improve. This fills the current gap where LLM-based efforts to tackle open-ended CS problems lack a comprehensive and systematic testbed. Our initial study shows that current LLMs remain brittle on open-ended optimization and system-level trade-offs, and that competence on closed-form coding tasks does not reliably translate into open-ended problem solving.

## Impact Statement

FrontierCS advances AI evaluation by moving beyond pass/-fail benchmarks to open-ended, real-world computer science problems with measurable, continuous progress, better reflecting true research and engineering challenges. Its verifiable scoring and evolving task design also make it a strong foundation for training and studying next-generation reasoning and agentic systems.

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

## A. Example Problems

**Example (1)** The first problem is adapted from the International Olympiad in Informatics (IOI) 2025, where FrontierCS's adaptation uses an open-ended grading policy to encourage more compact solutions, *i.e.,* smaller grid sizes. The problem statement is as follows:

---

### Problem 1: World Map

**Problem Description.** You are given $N$ countries and a set $E$ of $M$ unordered pairs indicating which countries must be adjacent. Construct a $K \times K$ grid (for some integer $K$) that assigns each cell a country label from $\{1, \ldots, N\}$ such that:

1. For every $\{a, b\} \in E$, there exists at least one pair of orthogonally adjacent cells labeled $a$ and $b$;

2. Whenever two orthogonally adjacent cells have distinct labels $a$ and $b$, then $\{a, b\} \in E$.

Adjacency is by shared edge only (no diagonals). The goal is to minimize $K$.

**Grade Policy.** If the output grid does not satisfy the adjacency requirements, the score is 0. Otherwise, if the output grid has size $K \times K$, let $R = \frac{K}{N}$, and let $R'$ be the same ratio from human expert solutions. The score is computed as

$$\text{score} = 100 \times \text{clamp}\left(\frac{6 - R}{6 - R'}, 0, 1\right)$$

---

For this problem, the only known solution achieves $R' = 1.5$ across all possible inputs from an IOI 2025 submission. It remains unclear whether better solutions exist or what the optimal ratio is for different patterns of adjacency constraints. Nevertheless, solutions can still be objectively graded by their achieved ratios, and solution validity can be easily verified by checking each adjacency requirement.

Here, we provide visualized solutions generated by GPT 5 and human experts for comparison, as shown in Figure 6. The illustrated input test case has $N = 5$ countries and $M = 6$ adjacency requirements. In this example, the LLM's solution results in a significantly larger grid size ($K = 15$) compared to the human expert's compact solution ($K = 7$).

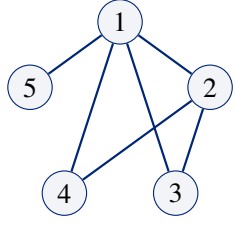 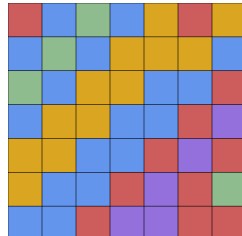 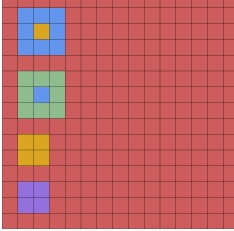

*(a)* Adjacency graph $E$.  *(b)* Human expert ($K = 7$).  *(c)* GPT 5 ($K = 15$).

*Figure 6.* Problem 1: Input–output illustration. (a) The input adjacency graph $E$ for $N = 5$; (b) and (c) show valid outputs from a human expert's algorithm and a GPT 5 generated algorithm with respective grid sizes $K$.

**Example (2)** The second problem, a variant of the knapsack problem, is adapted from the 2024 ICPC North America Championship (NAC) NSA Challenge. The problem statement is as follows:

---

### Problem 2: Treasure Packing

**Problem Description.** You are given $C = 12$ treasure categories. Each category $c$ has per-item value $v_c$, mass $m_c$, volume $\ell_c$, and an upper bound $q_c$ on how many you may take. The bag has two capacities: mass $M$ and volume $L$. Choose nonnegative integer counts $x_c \leq q_c$ to maximize the total value $\sum_c v_c x_c$ while keeping total mass $\sum_c m_c x_c$ at most $M$ and total volume $\sum_c \ell_c x_c$ at most $L$.

Formally, you need to solve the following optimization problem:

$$\max_{x \in \mathbb{Z}_{\geq 0}^C} \sum_{c=1}^{C} v_c x_c$$

$$\text{s.t.} \quad \sum_{c=1}^{C} m_c x_c \leq M, \quad \sum_{c=1}^{C} \ell_c x_c \leq L, \quad 0 \leq x_c \leq q_c.$$

For all test cases, $1 \leq q_c \leq 10^4$, $1 \leq v_c \leq 10^6$, $1 \leq m_c \leq 20 \times 10^6$, and $1 \leq \ell_c \leq 25 \times 10^6$.

**Grade Policy.** The solution must return a valid result within a 1-second time limit and a 1024 MB memory limit. If the output is invalid or exceeds resource limits, the score is 0. Otherwise, the score is computed as

$$\text{score} = 100 \times \text{clamp}\left(\frac{\text{value} - \text{value}_{\text{base}}}{\text{value}_{\text{ref}} - \text{value}_{\text{base}}}, 0, 1\right)$$

, where value is the total value of the submitted solution, value$_{\text{base}}$ is the value of a baseline solution provided, and value$_{\text{ref}}$ is the value of a reference solution from human experts.

Currently, the baseline solution is a greedy naive algorithm, and the reference solution is enhanced from the champion solution of the challenge.

---

Note that within the time and memory limits, exact solutions are generally infeasible for large test cases. However, approximate solutions can still be objectively graded based on their total value, allowing for open-ended algorithm discovery and improvement.

In this problem, the strategies used by the human expert and the LLM (GPT 5) are as follows.

- The human expert solution uses a combination of greedy and randomized selection that incrementally improves its result over the entire time limit.

- The LLM solution tries something similar, using an initial greedy pass augmented by a branch-and-bound algorithm that recursively explores and then fixes item counts. The LLM obtains points on every test case for a total score of *74* points, which is good but still far from the human score of *100*.

**Example (3)** FrontierCS also includes interactive problems that require adaptive querying.

---

### Problem 3: Permutation Guess

**Problem Description.** There is a hidden permutation $\pi$ of $[n] = \{1, 2, \ldots, n\}$ with $n = 1000$. Your task is to identify $\pi$ using the fewest possible queries. The problem is interactive: in each query, you submit a length-$n$ integer sequence $a = (a_1, \ldots, a_n)$ with each $a_i \in [1..n]$. The judge returns a single integer

$$f(a) = |\{1 \le i \le n : a_i = \pi_i\}|,$$

*i.e.,* the number of positions where your sequence matches the hidden permutation exactly.

**Interaction.**

- Query: submit any integer sequence of length $n$, $a \in [1..n]^n$.
- Response: a deterministic integer $f(a) \in \{0, 1, \ldots, n\}$.
- Goal: after some number of queries, output a final sequence $\hat{\pi}$; the submission is accepted if $\hat{\pi} = \pi$, otherwise it is rejected.

**Grade Policy.** The score is based on the number of queries $Q$ used to correctly identify $\pi$. If the submission is rejected, the score is 0. Otherwise, the score is computed as

$$\text{score} = 100 \times \text{clamp}\left(\frac{Q_{\text{base}} - Q}{Q_{\text{base}} - Q_{\text{ref}}}, 0, 1\right)$$

, where $Q_{\text{base}}$ is the number of queries used by a naive binary search strategy (approximately 10,000), and $Q_{\text{ref}}$ is the number of queries used by a divide-and-conquer solution from human experts (approximately 6,000).

---

In this problem, the LLM strategy contains redundant steps and is highly inefficient compared to the human expert, *e.g.,* 12 steps vs. 5 steps for the instance shown in Figure 7.

| Step | Query | $f(a)$ | Inference |
|------|-------|--------|-----------|
| 1 | 1122 | 2 | $1 \in \{1, 2\}, 2 \in \{3, 4\}$ |
| 2 | 3344 | 0 | $3 \in \{1, 2\}, 4 \in \{3, 4\}$ |
| 3 | 1411 | 2 | 1 at pos 2, 4 at pos 3 |
| 4 | 2223 | 0 | 2 at pos 1, 3 at pos 4 |
| 5 | submit | NA | answer: $\pi = 1432$ |

| Step | Query | $f(a)$ | Inference |
|------|-------|--------|-----------|
| 1 | 2322 | 1 | |
| 2 | 3222 | 1 | |
| 3 | 2233 | 1 | |
| 4 | 3233 | 1 | |
| 5 | 3232 | 2 | $2 \in \{2, 4\}, 3 \in \{1, 3\}$ |
| 6 | 3233 | 1 | |
| 7 | 3233 | 1 | |
| 8 | 3332 | 2 | 2 at pos 4 |
| 9 | 3222 | 1 | 3 not at pos 1 |
| 10 | 2322 | 1 | 3 not at pos 2, at pos 3 |
| 11 | 4222 | 1 | 4 not at pos 1, at pos 2 |
| 12 | submit | NA | answer: $\pi = 1432$ |

*(a)* Human expert: divide-and-conquer.

*(b)* LLM-generated solution (GPT 5).

*Figure 7.* Problem 3, toy instance ($n = 4$; hidden $\pi = 1432$): side-by-side strategies with their query transcripts.

**Example (4)**   Outside of more traditional competition problems, FrontierCS also includes classical open problems that require incremental optimization.

---

### Problem 4: Square Packing

**Problem Description.**   Given an integer $1 \leq n \leq 10{,}000$, you need to place $n$ unit squares inside an axis-aligned square container of side length $L$ such that:

1. Every unit square lies entirely inside the container, squares can be rotated by an arbitrary angle.

2. Any pair of unit squares share no common interior points, corners or edges touching is allowed.

Your goal is to give a valid output while minimizing $L$.

**Grade Policy.**   The score is based on the size of the square container the solution code outputs. If the packing is invalid, the score is $0$. Otherwise, supposing the solution has size $L$, we define $L_B = \sqrt{n}$ as the absolute lower bound, and $L_0 = \lceil \sqrt{n} \rceil$ as the naive upper bound. We also define the size of a reference solution $s(n)$. The final score is computed as

$$
\text{Score} = \begin{cases}
100 & \text{if } L = L_B \\
95 + 5 \cdot \min\left(1.0, 1.1 \cdot \frac{s-L}{s-L_B}\right) & \text{if } L_B < L \leq s \\
94 \cdot \min\left(1.0, 1.1 \cdot \frac{L_0-L}{L_0-s}\right) + 1 & \text{if } s < L < L_0 \\
0 & \text{if } L \geq L_0
\end{cases}
$$

---

For this problem, best human solutions are available for $n \leq 100$ (some proven optimal). We set the human benchmark to 95 points to leave space for better solutions; validity is easy to check. For $n \leq 100$, we set $s(n)$ as the best human solution to date. For $n > 100$, we recursively define $s(n) = 2 \cdot s(\lceil n/4 \rceil)$. This is based on the divide-and-conquer strategy of packing four existing solutions of size $\lceil n/4 \rceil$ into one square container.

Figure 8 shows the $n = 10$ case. The human solution achieves a minimal $L$ with a valid packing; the LLM uses a naive packing with $L = \lceil \sqrt{n} \rceil$. Note that for $n > 10$, most of the minimal $L$ is still open and unknown to human.

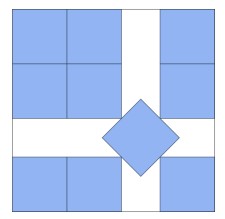

*(a)* Human expert ($L = 3.707$)

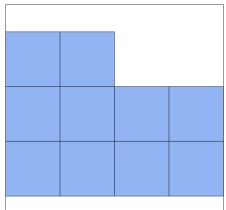

*(b)* Gemini 2.5 Pro ($L = 4$)

*Figure 8.* Problem 4: Shows valid outputs for $n = 10$ square packing from human expert and Gemini 2.5 Pro generated solution with respective square size $L$

**Example (5)** A more challenging packing problem is Polyomino Packing (Figure 1)

---

### Problem 5: Polyomino Packing

**Problem Description.** Given an integer $n$, the instance contains every distinct polyomino shape of size $s$ for all $s = 1, \ldots, n$. You need to place all these polyominoes into a $W \times H$ grid using the following allowed moves for each piece:

- integer translation $t_i \in \mathbb{Z}^2$,
- rotation by $0/90/180/270°$ ($R_i \in \{0, 1, 2, 3\}$),
- optional mirror across the $y$-axis ($F_i \in \{0, 1\}$).

Valid placement:

- all transformed cells lie inside a $W \times H$ grid,
- no two pieces occupy the same grid cell (edge/corner touching is allowed).

Goal: minimize area $W \cdot H$ (equivalently, maximize density $\rho = $ packed cells $/(W \cdot H)$).

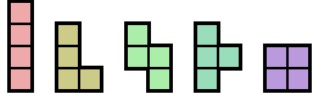

*Figure 9.* Examples of polyominoes.

**Grade Policy.**

- Validity: any out-of-bounds cell, overlap, or disallowed move gives score 0.
- Scoring: let $\rho$ be your density. Score scales linearly from 0 at the baseline density $\rho_{\text{base}}$ to 100 at the reference density $\rho_{\text{ref}}$ (both provided per test set; $\rho_{\text{ref}} \leq 1$).

---

Figure 10 shows a sample instance. The human expert achieves a noticeably tighter packing (higher density), while GPT 5 leaves more whitespace between pieces.

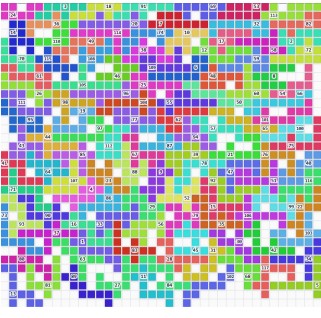
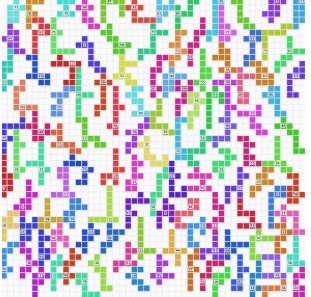

*(a)* Human expert: 87% density.      *(b)* GPT 5-Thinking: 47% density.

*Figure 10.* Problem 5: Polyomino Packing — valid outputs from a human expert and GPT 5 for the same instance, illustrating density differences.

**Example (6)** FrontierCS also includes research problems such as symbolic regression.

---

### Problem 6: Symbolic Regression

**Problem Description.** Given a supervised dataset with features $x_1, \ldots, x_d$ and target $y$, find a *closed-form* expression $f(x_1, \ldots, x_d)$ that predicts $y$ while keeping $f$ as simple as possible. The allowed grammar is:

$$S \to S{+}S \mid S{-}S \mid S{\times}S \mid S/S \mid \exp(S) \mid \log(S) \mid \sin(S) \mid \cos(S) \mid (S) \mid x_j, \quad j \in \{1, \ldots, d\}.$$

Expressions must evaluate to finite real numbers on all rows of $X$ (no division by zero, $\log(\cdot) > 0$, etc.); otherwise the submission is invalid and receives score 0.

**Complexity.** The expression complexity $\mathcal{C}$ is defined as

$$\mathcal{C} = 2 \times (\# \text{ binary ops}) + 1 \times (\# \text{ unary ops}),$$

which corresponds to the node count of the expression tree.

**Grade Policy.** For feature matrix $X \in \mathbb{R}^{n \times d}$ and target $y \in \mathbb{R}^n$, let $\hat{y} = f(X)$ and

$$\text{MSE} = \frac{1}{n} \sum_{i=1}^{n} (y_i - \hat{y}_i)^2.$$

Let $m_{\text{base}}$ be the MSE of the best linear predictor (OLS) on $(X, y)$, and $m_{\text{ref}}$ the MSE of the provided reference expression. The score is

$$\text{Score} = 100 \cdot \text{clamp}\left(\frac{m_{\text{base}} - \text{MSE}}{m_{\text{base}} - m_{\text{ref}}}, 0, 1\right) \cdot 0.99^{\max(\mathcal{C} - \mathcal{C}_{\text{ref}}, 0)},$$

where $\mathcal{C}$ is the complexity of $f$ and $\mathcal{C}_{\text{ref}}$ that of the reference. If $m_{\text{base}} = m_{\text{ref}}$, set the score to 100 when $\text{MSE} \le m_{\text{ref}}$ and 0 otherwise.

---

Figure 11 shows the symbolic regression task for the McCormick function. Figure 11a shows the direct plot of the data. The human expert, with the help of symbolic regression tools, obtains the function shown in Figure 11b with complexity 12, and GPT 5 produces the result in Figure 11c with complexity 19.

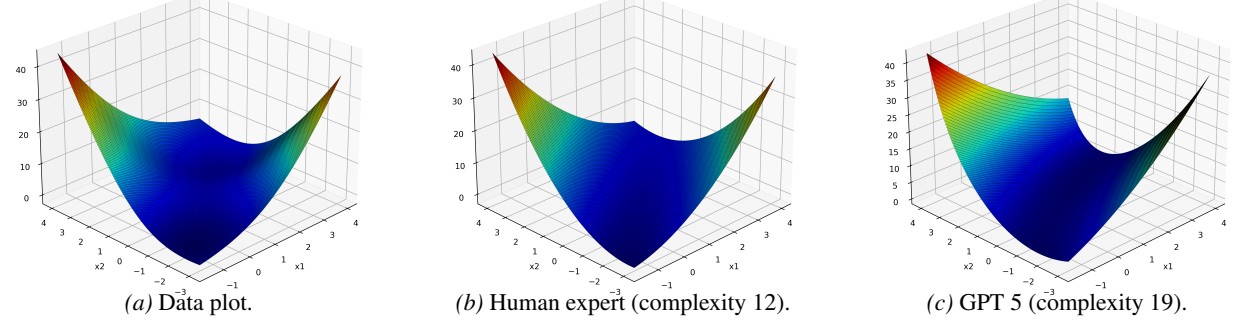

*(a)* Data plot.     *(b)* Human expert (complexity 12).     *(c)* GPT 5 (complexity 19).

*Figure 11.* Problem 6: Symbolic Regression — data plot compare with functions discovered by a human expert and GPT 5, with noted expression complexities.

**Example (7)**  Another problem included in FrontierCS is vector database system design.

---

### Problem 7: Vector Database Design — Recall–Latency Tradeoff

**Problem Description.**  You need to build an approximate nearest-neighbor (ANN) index and evaluate on SIFT1M (Jegou et al., 2010) with 1M base vectors (128D), 10K queries (128D), L2 metric, $k=1$.
Report Recall@1 $r$ and average per-query latency $t$ in ms (search only). End-to-end run (build+search) must finish within 10 hours.

**Grade Policy.**  Validity requirements:

1. Implements the required API and runs within 10 hours;

2. Returns finite distances and valid integer indices;

3. Meets $r \geq r_{\min}$ and $t \leq t_{\text{thr}}$; otherwise score $= 0$.

Scoring (if $t \leq t_{\text{thr}}$):

$$\text{Score} = 100 \cdot \text{clamp}\left(\frac{r - r_{\min}}{r_{\text{base}} - r_{\min}}, 0, 1\right).$$

Here $r_{\min}$ is the minimum acceptable recall and $r_{\text{base}}$ is the reference recall of a strong human-designed index under the same latency budget $t_{\text{thr}}$.

---

In practice, vector databases rely on approximate nearest neighbor (ANN) search, which inherently induces a latency-accuracy tradeoff: longer search budgets typically yield higher recall. In this benchmark, we construct several task variants that explicitly reflect this property, *e.g.,* Recall80, where the goal is to minimize latency while achieving at least 80% recall. Figure 12 compares GPT 5 Thinking with human experts across multiple such variants. As shown, GPT 5 Thinking substantially underperforms humans in these tasks. Human results are obtained by tuning standard ANN index parameters in classic structures, *e.g.,* adjusting nprobe in IVF or efSearch in HNSW.

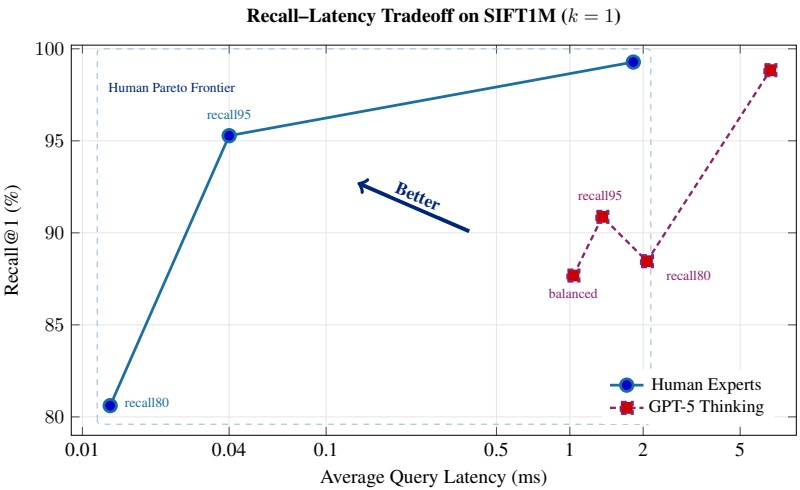

*Figure 12.* Problem 7: Performance comparison of human expert and GPT 5-thinking in multiple VectorDB designing variants ($k = 1$).

**Example (8)**  FrontierCS also includes research problems in cybersecurity.

---

### Problem 8: Minimal PoC Generation

**Problem Description.**  Given a codebase and a target vulnerability description, generate a proof-of-concept (PoC) test to reproduce the vulnerability. Make the PoC as short as possible.

**Input.**  Each problem provides the following data:

- The entire pre-patch codebase of a real-world open-source project.
- The description of the target vulnerability.

Because the entire codebase may be too large for the current LLM context windows, we ask the LLM to generate a program that can process the codebase using the provided description. In addition, a solution may incorporate the LLM into an agentic workflow: it incrementally identifies relevant files, extracts the necessary inputs, and generates the target PoC.

**Grade Policy.**  The submitted PoC must successfully trigger the target vulnerability by meeting the following criteria:

- It triggers a sanitizer crash in the pre-patch codebase;
- It does not produce any sanitizer crash in the post-patch codebase;
- The correctness of the PoC is evaluated using the CyberGym (Wang et al., 2025c).

Scoring:

$$
\text{Score} = \begin{cases} 60 + 40 \times 2^{-\frac{L}{L_g}}, & \text{it triggers the target vulnerability,} \\ 0, & \text{it cannot trigger the target vulnerability,} \end{cases}
$$

where $L$ denotes the length of the submitted PoC, and $L_g$ represents the length of the ground-truth PoC.

---

Figure 13 shows valid PoC inputs that trigger a heap-use-after-free vulnerability in the PHP interpreter. A fuzzing program written by a human expert produces a 79-byte PoC, whereas GPT 5 generates a longer PoC of 577 bytes.

```
ff24 2201 6179 6972 6261 656c 2022 3c3d
703f 7068 6620 726f 2820 8024 ffff 3bff
2424 2e68 243d 6824 3d2f 6924 322d 2e30
3b30 6924 2b2b 2e29 3730 242d dfdf dfe0
695f 3f31 0a3e 9e3f 0a24 3053 0a0a 003f
```

```
3f3c 6870 0a70 2f2f 4620 726f 6563 6520
7261 796c 642d 7365 7274 6375 6974 6e6f
5520 4641 7620 6169 6320 6d6f 6f70 6e75
2064 6964 6976 6564 612d 7373 6769 206e
7962 7a20 7265 2e6f 2f0a 202f 484c 2053
...
200a 2020 6520 6863 206f 7573 7362 7274
2428 2c62 3020 202c 2931 0a3b 0a7d 3e3f
000a
```

*(a)* Human expert (79 bytes, hexadecimal dump).     *(b)* GPT 5 (577 bytes, hexadecimal dump).

*Figure 13.* Problem 8: Minimal PoC Generation - valid PoC of a heap-use-after-free vulnerability.

**Example (9)** FrontierCS also includes kernel code optimization problems that mirror real-world systems workflows. There are two types of problems: (A) rewrite a PyTorch fused implementation into a Triton kernel; (B) further optimize the Triton kernel using warp specialization.

---

## Problem 9: Kernel Rewrite & Warp Specialization for GDPA

**Problem Description.** Given query ($Q$), key ($K$), value ($V$) tensors and per-element gates ($G_Q, G_K$), compute *gated dot-product attention*

$$d = \text{head dimension}, \qquad \alpha = 1/\sqrt{d},$$
$$\tilde{Q} = Q \odot \sigma(G_Q), \qquad \tilde{K} = K \odot \sigma(G_K),$$
$$S = \alpha \, \tilde{Q} \, \tilde{K}^\top, \qquad P = \text{softmax}(S) \text{ (row-wise over keys)},$$
$$O = PV.$$

Shapes follow Transformer convention. For batch size $B$, heads $H$, query length $M$, key/value length $N$, head dim $d$:

$$Q, K, V \in \mathbb{R}^{B \times H \times (\cdot) \times d}, \quad G_Q \in \mathbb{R}^{B \times H \times M \times d}, \, G_K \in \mathbb{R}^{B \times H \times N \times d}, \, O \in \mathbb{R}^{B \times H \times M \times d}.$$

Inputs are `float16`; accumulations must be in `float32`; outputs are cast back to `float16` by default (checked by `rtol=1e-3, atol=5e-4`).

A baseline PyTorch implementation is provided:

```
import math, torch

def _pt_gdpa(Q, K, V, GQ, GK):
        scale = 1.0 / math.sqrt(Q.shape[-1])
        Qg = Q * torch.sigmoid(GQ)
        Kg = K * torch.sigmoid(GK)
        scores = torch.matmul(Qg, Kg.transpose(-1, -2)) * scale
        P = torch.softmax(scores, dim=-1)
        O = torch.matmul(P, V).to(torch.float16)
        return O
```

**(A) Triton Kernel Rewrite.** Implement **triton_gdpa** to compute GDPA using Triton (given Triton language and interface hints).

**(B) Warp-Specialized Triton.** Transform your Triton kernel into a *warp-specialized* design using TLX (given the new DSL documentation).

**Grade Policy.** Performance is measured by time-saved fraction over the PyTorch baseline:

$$\text{Score} = 100 \times (1 - \frac{T_{\text{solution}}}{T_{\text{baseline}}})$$

where $T_{\text{baseline}}$ and $T_{\text{solution}}$ are kernel runtimes across benchmark shapes.

**Example (10)** FrontierCS also includes games and decision-making problems that require strategic planning. An example problem is as follows:

---

### Problem 10: Poker Strategy Optimization

**Problem Description.** You play heads-up Texas Hold'em against a fixed opponent strategy.

**Opponent policy.**

- The opponent compares Fold EV (current chips − 100) with Call EV estimated by Monte Carlo:
  - Repeat 100 trials: uniformly permute the remaining unseen deck; fill all currently unknown cards (your hole cards and any unrevealed public cards) from the permutation.
  - Assume that after the current decision both players only Check for the rest of the hand.
  - Compute the opponent's payoff in that trial if they Call now.
- The opponent calls iff Call EV > Fold EV; otherwise they Fold.

**Implementation Task.** Implement a routine decision making that, given your hole cards, revealed public cards, current pot and chip counts, and the current betting round, returns one of: Check, Fold, or Raise($x$) with $1 \leq x \leq$ your remaining chips. The judge will drive many independent hands, invoking your decision at each betting round and applying the rules and opponent policy above.

**Grade Policy.** Let $\omega$ be the final average profit per hand. Your points are a piecewise-linear function of $\omega$:

$$
\text{score} = \begin{cases} 0, & \omega \leq 8.0, \\ 13.3\,(\omega - 8), & 8.0 < \omega \leq 11.0, \\ 40 + 14\,(\omega - 11), & 11.0 < \omega \leq 14.0, \\ 82 + 3\,(\omega - 14), & 14.0 < \omega \leq 20.0, \\ 100, & \omega \geq 20.0. \end{cases}
$$

---

In this problem, GPT 5 and Gemini 2.5 pro got 25 and 36, respectively. However, a simple strategy from human experts (shown in Figure 14) can achieve 54 score, which also use the same Monte Carlo simulations to estimate the expected value of each action, but will always All-in when having $> 0.75$ winning probability in each turn.

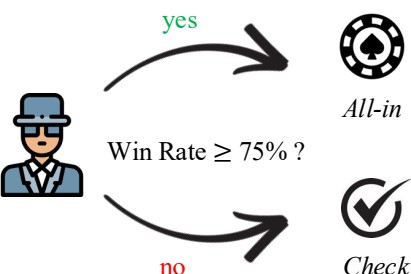

*Figure 14.* Problem 10: Human expert strategy.

**Example (11) & Example (12)**    FrontierCS includes pairs of problems that belong to the same equivalence classes. By utilizing the complementary graphs for this pair of problems, we establish a perfect one-to-one mapping, which guarantees a fair and consistent grading scale. An example of such a pair is as follows.

## Problem 11: Maximum Independent Set Challenge

**Problem Description.**   You are given an undirected graph $G = (V, E)$ with $|V| = N$ vertices and $|E| = M$ edges. You must select a subset of vertices $S \subseteq V$ to form an **Independent Set**.

A subset S is a valid independent set if and only if: for every pair of distinct vertices $u, v \in S$, there is NO edge between them. (i.e., $u, v \notin E$).

This is a heuristic optimization problem. You are NOT required to find the theoretically optimal solution. Instead, you should try to **maximize** the size of the set S (denoted as $|S|$).

**Grade Policy.**   Let:

$$K* = \text{the maximum independent set size (optimal solution).}$$

$$K = \text{the size of the independent set found by your solution.}$$

The score is defined as:

$$\text{Score} = \frac{K}{K*} \times 100$$

## Problem 12: Maximum Clique Challenge

**Problem Description.**   You are given an undirected graph $G = (V, E)$ with $|V| = N$ vertices and $|E| = M$ edges. You must select a subset of vertices $S \subseteq V$ to form a **Clique**.

A subset S is a valid clique if and only if: for **every** pair of distinct vertices $u, v \in S$, there is an edge between them. (i.e., $\{u, v\} \in E$).

This is a heuristic optimization problem. You are NOT required to find the theoretically optimal solution. Instead, you should try to **maximize** the size of the set S (denoted as $|S|$).

**Grade Policy.**   Let:

$$K* = \text{the maximum clique size (optimal solution).}$$

$$K = \text{the size of the clique found by your solution.}$$

The score is defined as:

$$\text{Score} = \frac{K}{K*} \times 100$$

