# OpenReview forum: "FrontierCS: Evolving Challenges for Evolving Intelligence"
_ICML.cc/2026/Conference — ICML 2026 regular_

### Official Review · Reviewer_AoRe · 2026-03-11

**Soundness:** 3
**Presentation:** 3
**Significance:** 3
**Originality:** 4
**Overall Recommendation:** 5
**Confidence:** 3

**Summary:**

This paper offers a benchmark for open-ended problems for which no known deterministic solutions exist in practice (e.g. NP Hard variants). Scoring is determined using a trivial baseline solution and human expert reference solution. The benchmark is tested on several SOTA models and findings include lag between models and human experts, brittleness of agents, and the failure of reasoning budgets to close the gap.

**Compliance With Llm Reviewing Policy:**

Affirmed.

**Key Questions For Authors:**

none

**Limitations:**

yes

**Strengths And Weaknesses:**

This paper makes a genuine contribution to the field by focusing on a) NP Hard variants b) with no optimal solution using c) partial scoring and d) offering a comparison ceiling of human expert reference. Their findings are not new, so they are confirmatory and not surprising. However, as models advance, this approach could offer ways to assess SOTA advances. The one weakness is that scoring remains (perhaps necessarily) somewhat fuzzy, at least as it is explained here, although the authors do provide a sample example in the appendix.

---

> ### Author Rebuttal · Authors · 2026-03-26
>
> ### 1. Clarification on Scoring
>
> We thank the reviewer for the positive assessment and helpful feedback. We provide concrete examples in the paper to illustrate the details of our scoring design. Since tasks differ significantly in nature and difficulty, we adopt **task-specific scoring criteria** tailored to each problem.
>
> This design helps prevent cases where models achieve large score gains through trivial improvements, and instead ensures that the evaluation reflects **meaningful algorithmic progress**. We will further clarify this aspect in the final version.

---

> > ### Author Rebuttal · Reviewer_AoRe · 2026-04-03
> >
> > The normalization and methodology described in your rebuttal to other reviewers somewhat addresses my concern about scoring transparency. However, this detail needs to be in the paper, not just the rebuttal. I'd also appreciate clarification on how the benchmark maintains comparability over time as human references are updated.

---

> > > ### Author Response · Authors · 2026-04-08
> > >
> > > Thank you for this valuable feedback. we fully agree that these details should be included in the main paper after the rebuttal. We would like to further clarify that human reference solutions are used only as normalization anchors, not as absolute ground truth. We score model outputs relative to human solutions, as illustrated in the examples shown after page 10. Therefore, when the human solutions become stronger, the model scores will correspondingly decrease.

---

### Official Review · Reviewer_vGm6 · 2026-03-12

**Soundness:** 3
**Presentation:** 3
**Significance:** 3
**Originality:** 3
**Overall Recommendation:** 4
**Confidence:** 3

**Summary:**

The paper introduces FrontierCS, a novel benchmark consisting of 240 open-ended computer science problems. Unlike traditional benchmarks that evaluate binary pass-or-fail correctness on problems with known optimal solutions, FrontierCS targets intractable or unsolved tasks where quality is evaluated objectively on a continuous spectrum. The benchmark spans both algorithmic challenges and practical research problems in areas such as OS, HPC, DB, PL, AI, and Security. Submissions are evaluated via task-specific deterministic scoring of executable programs under resource limits. For example, a polyomino packing task is scored based on the packing density achieved by the model's program. The authors evaluate state-of-the-art LLMs, demonstrating that they lag significantly behind human experts. Furthermore, the empirical results show that models suffer from formulation sensitivity (failing to recognize equivalence classes) and that agentic frameworks overfit when applied to these tasks.

**Compliance With Llm Reviewing Policy:**

Affirmed.

**Key Questions For Authors:**

1. How computationally expensive is the automatic evaluator to run over the full suite of 240 problems for a single model evaluation, particularly for the systems and HPC research tasks?
2. Could you elaborate on how the scores from disparate tasks with different scales (e.g., packing density percentages vs. execution times) are normalized or aggregated into a summary metric for the benchmark?
3. In the comparison between human experts and models, how much time and execution feedback (e.g., access to a compiler, local testing, or iterative debugging) were human experts allowed to use compared to the models' one-shot generation?

**Limitations:**

yes

**Strengths And Weaknesses:**

Strengths:

- **Originality**: The benchmark represents a highly original shift from traditional closed-form coding evaluations (which models are currently saturating) to open-ended, verifiable optimization problems.
- **Significance**: Evaluating the ability of models to navigate complex trade-offs in real-world CS research is a critical and timely contribution to the community. The inclusion of diverse domains like systems optimization and cybersecurity significantly broadens the impact of the work.
- **Soundness**: The methodology is highly robust. The inclusion of parametric problem generators to prevent data leakage and overfitting, paired with expert reference solutions and deterministic automated evaluators, ensures the benchmark is reproducible and objective.
- **Insights**: The paper provides valuable empirical observations, specifically highlighting that current models fail to normalize equivalent formulations of a problem and that agentic systems struggle with overfitting when trained on subsets of test cases.

Weaknesses:

- **Evaluation Setting Alignment**: The models are primarily evaluated in a one-shot setting without execution feedback, debugging, or iterative refinement. Since open-ended optimization typically involves heavy iterative debugging by human experts, comparing one-shot model outputs directly to human expert performance might not represent a fully fair baseline.
- **Score Aggregation Clarity**: It is not entirely clear from the high-level description how diverse continuous metrics (e.g., density in polyomino packing vs. execution time in Triton kernel optimization) are normalized and aggregated into a reliable global score for overarching model comparison.

---

> ### Author Rebuttal · Authors · 2026-03-27
>
> ### 1. Computational Cost of Evaluation
>
> The computational cost varies significantly across the two tracks:
> - **Algorithmic problems**: Evaluation is lightweight. Each solution is executed on standard hardware with the specified time limit. Evaluating all algorithmic problems for a single model takes approximately **4-6 hours** on a single machine, including compilation, execution, and scoring.
> - **Research problems**: These are more resource-intensive. Some tasks (e.g., vector database design, kernel optimization) require GPU instances or specialized hardware. We use SkyPilot (Yang et al., 2023) to manage cloud infrastructure, leveraging spot instances for cost efficiency. Evaluating all research problems for a single model takes approximately **12-24 hours** using distributed cloud resources, at an estimated cost of **$50-100** per full evaluation run.
>
> We will add these details to the paper.
>
> ---
>
> ### 2. Score Normalization and Aggregation
>
> We use a two-level approach:
> 1. **Task-level normalization**: Each task's raw metric (e.g., packing density, execution time, number of queries) is mapped to a 0-100 score using the formula: `score = 100 * clamp((metric - baseline) / (reference - baseline), 0, 1)`. The baseline is a trivial solution and the reference is the human expert solution. This ensures all tasks are on a common scale, where 0 = trivial baseline performance and 100 = human expert performance.
> 2. **Cross-task aggregation**: We report the average normalized score across tasks (Score@k, Avg@k). Additionally, we compute **Elo ratings** using pairwise comparisons on each problem, which inherently accounts for difficulty differences and avoids the pitfalls of averaging heterogeneous metrics.
>
> The Elo system is our recommended metric for model comparison, as it is robust to the scale differences across tasks. The normalized scores are complementary and provide interpretable per-task diagnostics.
>
> ---
>
> ### 3. Human Expert Conditions vs. Model Conditions
>
> This is an important distinction that we acknowledge:
> - **Human experts** developed reference solutions with access to execution feedback, debugging tools, and iterative refinement. They had no strict time limit (typically 2-8 hours per problem).
> - **Models** are evaluated in a single-round, zero-shot setting without execution feedback.
>
> This asymmetry is intentional: the human reference represents a **strong performance ceiling**, not a controlled experimental comparison. The primary goal is to measure how far models can go on intrinsic algorithmic reasoning, and the human reference provides an interpretable upper bound. We acknowledge this in Section 6 and note that future work will include controlled comparisons where models also have access to iterative execution, making the comparison more direct. Our preliminary agentic results (Section 5.4) represent a first step in this direction.

---

### Official Review · Reviewer_gavK · 2026-03-12

**Soundness:** 3
**Presentation:** 3
**Significance:** 2
**Originality:** 2
**Overall Recommendation:** 3
**Confidence:** 4

**Summary:**

-	This paper introduces a new benchmark of computer science optimisations problems whose solution is in the form of a program that has to find a solution to a problem that has not known optimal solution but its quality can be checked automatically. The authors show that for this benchmark, state-of-the-art AI models have some issues, so it is useful to guide further progress.

**Compliance With Llm Reviewing Policy:**

Affirmed.

**Final Justification:**

I looked at the other reviews, and the authors clarified some questions, but the reasons behind my score remain.

**Key Questions For Authors:**

QUESTIONS
-	How well-known are the collected problems or variations of existing problems to determine the possible contamination level? If this is known per task, how does the difference between humans and AI system correlate with this? (in any case the answer to this question is less informative as everything is biased by excluding the tasks for which AI was already better)
-	I don’t understand the third point of section 3.3. How is it that changing the human reference makes the problem more difficult? This is just a different normalisation of the score, but it doesn’t make it more difficult.

**Limitations:**

-	The authors mention the limitation that the models cannot run the algorithm (program) they are writing. I assume humans can’t do that either.

**Strengths And Weaknesses:**

STRENGTHS
-	More challenging benchmarks are needed, especially in coding and mathematics.
-	Good coverage of the state of the art.
-	Good characterisation of the problems in a useful taxonomy.
-	The methodology for collecting the problems is well designed and curated.
-	The qualitative analysis in section 5 is insightful.
WEAKNESSES
-	It’s not fully clear how the time complexity of the proposed solution program affects the metrics used to compare humans and LLMs. With unbounded time, if both humans and AI know, they would code a complete yet inefficient algorithm. Are they told how computational complexity is considered? It’s only on page 4 that I read that problems “are solved under time and memory limits”, but not further details are given. This is superimportant: what these limits are and whether/how they are told to humans and AI systems.
-	Some degree of difficulty assessment of each problem would have been useful, in terms of the scores, to be able to combine the results of several tasks in a more meaningful way. This is actually an opportunity of this benchmark, as they can gauge the difficulty, what I mean is some kind of calibration across different tasks.
-	I found that for the problems in 3.1 “the human reference solution is significantly stronger than the best model’s performance”. This is good for making the benchmark challenging, but makes it clearly biased for answering the question on whether AI systems are still lagging behind humans in open CS problems, which is one of the “findings” of the paper. This should be corrected and clearly highlighted. Unfortunately, they should have kept some of them, and note the proportion. This would have allowed for some normalisation to counteract the bias, and make the measurements more meaningful. For now, the benchmark is useful for encouraging progress in AI systems, but not as a way to compare how good AI is compared to humans.
-	They say that they “primarily evaluate solutions using …, rather than computational efficiency”, but if there’s a limit in steps this is already evaluated. In any case, the information about this (and whether the subjects are informed about it) is very imprecise.
-	For the problems in 3.2, time and memory are taken into account, but again, is this said? This is only clarified when reading about the “variants”. I then can understand that the “resource constraints” are told, finally, in all cases.
-	I miss a comparison with Humanity’s Last Exam, not only in terms of the problems, but methodology of collection, and also the same criterion of excluding problems for which AI systems are already better than humans.
-	The min-max normalisation against baseline and humans is not ideal, as the averages are still apples and oranges, but it’s a compromise, and is compensated by the use of the Elo score. However, I was expecting that this would allow for extrapolation, but my understanding is that doing worse than the baseline solution is not penalised, but at the same time the models get saturated if they do better than humans. I don’t know how frequently this happens now, but in the future it will report a highly distorted result of how close they are to humans. If I understood this correctly, this should be fixed.
-	Some sample problems are included in the appendix, which helps a lot, but I don’t see the actual prompts, so I now doubt again whether the humans and the models are told about the constraints in resources (time and space) in the prompt, and what score is being used. Did they have the same information, different information or no information at all. This is crucial. Without detailed information about this for all categories the paper shouldn’t be accepted.
-	There’s little information about the methodology for collecting the human baseline (setting, selection, demographics, etc.). This should be addressed, as well as the ethics of the human study (consent of participants, etc.)
-	The analysis of the reasoning effort is very preliminary and there should be some break downs about the cases where resources are relevant. It may be the case that more thinking leads to more or less efficient algorithms. This trade-off between the quality of the solution and the complexity of the algorithm can only be optimised if the thinking model this is what it has to optimise (in those cases where this is the case, or there’s a limit when there’s a limit).
- The title (except “CS”) is not very informative about the paper. I thought it was about continual learning. Clarifying this is a benchmark and it’s about computer science optimisation problems would help reviewers and future readers.

---

> ### Author Rebuttal · Authors · 2026-03-27
>
> ### 1. Time and Memory Limits: Transparency and Communication
>
> We appreciate this critical concern and agree that it deserves clearer presentation. To clarify:
>
> - **All problems have explicit time and memory limits**, which are communicated identically to both human experts and LLMs in the problem prompt. For algorithmic problems, these are specified as standard competitive-programming-style constraints (e.g., "1-second time limit, 1024 MB memory limit" as shown in Problem 2, Appendix A). For research problems, resource constraints are specified in the README (e.g., "end-to-end run must finish within 10 hours" for Problem 7).
> - **Both humans and models receive the same problem statement**, including all constraint information. The prompt provided to LLMs is the exact same text as the problem statement shown to human experts. We did not provide any additional hints or information to either party.
> - **Runtime is a feasibility constraint, not a scoring component** (unless explicitly stated otherwise). As stated in Section 3.1 (lines 231-234): "Unless explicitly specified otherwise, runtime is never part of the scoring, and together with the resource limits, this ensures that higher scores reflect better strategies rather than solutions that rely on excessive compute."
>
> ---
>
> ### 2. Difficulty Assessment and Cross-Task Calibration
>
> This is a thoughtful suggestion. We agree that a difficulty calibration across tasks would add value. In the current version, we partially address this through:
> - **Elo ratings** (Section 4.2), which inherently normalize for problem difficulty by computing pairwise win rates between models on each problem.
> - **Min-max normalization** against baseline and human reference solutions, which provides a common 0-100 scale. In the paper, we also provide several examples illustrating how we curve the scores differently based on the nature of each problem.
>
> ---
>
> ### 3. Bias from Excluding Tasks Where AI Outperforms Humans
>
> We appreciate this nuanced point, but we would like to respectfully clarify a misunderstanding. We do **not** exclude tasks based on whether AI outperforms humans. On the contrary, these tasks were originally designed to test humans, and we only adapted and curated them. The review criterion in Section 3.1 states that "the human reference solution is significantly stronger than the best model's performance"—this is a **quality control measure for the benchmark**, not a cherry-picking of results. The purpose is to ensure that the benchmark contains genuinely challenging problems that are not already saturated.
>
> This is analogous to the standard practice in benchmark design of selecting problems at an appropriate difficulty level. For example, HumanEval and MBPP were designed at a difficulty where early LLMs could not solve all problems; FrontierCS applies the same principle at a higher difficulty tier. We do not claim that AI lags behind humans on *all* CS problems—only on the class of open-ended, unsolved optimization problems that FrontierCS targets.
>
> ---
>
> ### 4. Min-Max Normalization and Score Saturation
>
> The reviewer raises a valid concern about the normalization scheme. To clarify:
>
> - A model performing **below** the trivial baseline receives a score of 0 (not negative). This is by design: the baseline represents the minimum acceptable quality, and we do not penalize below-baseline performance on a gradient since it does not meaningfully discriminate model capabilities.
> - A model performing **above** the human reference receives a score of 100 (capped). We acknowledge this creates a ceiling effect. However, in the current evaluation, this almost never occurs—no model consistently surpasses human experts. When models do improve to this level, our **update policy** (Section 3.3) prescribes raising the human reference or tightening constraints to maintain discriminativeness.
> - The **Elo rating system** (which operates on raw pairwise comparisons, not normalized scores) is specifically designed to avoid these normalization artifacts. We recommend Elo as the primary metric for model comparison and use the normalized scores for interpretability.
>
> We will add a discussion of these boundary cases and their expected impact as models improve.
>
> ---
>
> ### 5. Comparison with Humanity's Last Exam (HLE)
>
> Thank you for this suggestion. While both HLE and FrontierCS target frontier-level evaluation, they differ fundamentally in design philosophy: HLE tests whether models *know* the answer to hard questions, while FrontierCS tests whether models can design effective algorithms for problems where no one knows the optimal answer. These are complementary evaluation paradigms. We will add this comparison to the Related Work section.
>
> Regarding collection methodology, both benchmarks use expert curation. HLE's approach of excluding known-answer questions is analogous to our requirement that problems have no known optimal solution. Both aim to target the frontier of current capabilities.

---

> > ### Author Rebuttal · Reviewer_gavK · 2026-04-05
> >
> > I thank the authors for the response. It answers my questions but it also confirms my understanding was correct about the selection of questions and the normalisation. I've also seen the other reviews. I keep my score.

---

### Official Review · Reviewer_tLNH · 2026-03-12

**Soundness:** 4
**Presentation:** 4
**Significance:** 4
**Originality:** 4
**Overall Recommendation:** 4
**Confidence:** 4

**Summary:**

This paper introduces FrontierCS, a benchmark of 240 open-ended computer science problems (172 algorithmic, 68 research) designed to evaluate LLMs on tasks where no known optimal solution exists but solution quality can be objectively and deterministically scored on a continuous scale. Problems span optimization, constructive, and interactive algorithmic categories as well as research domains (OS, HPC, AI, DB, PL, SE, Security). The benchmark is curated through a multi-stage proposal-implementation-review pipeline involving competitive programming experts and CS PhD students. The authors evaluate seven frontier reasoning models (GPT-5 variants, Gemini 2.5/3.0 Pro, Grok 4, DeepSeek 3.2) and find that all models significantly underperform human experts (e.g., ~33 vs. ~87 on algorithmic problems). Additional analyses reveal that increasing reasoning budgets yields diminishing or negative returns, models are prone to "micro-optimization traps," they are sensitive to equivalent problem formulations, and agentic evolution frameworks like OpenEvolve overfit when trained on partial test subsets.

**Compliance With Llm Reviewing Policy:**

Affirmed.

**Key Questions For Authors:**

1. How were the human expert reference solutions produced? Were the problem setters also the ones providing reference solutions, and if so, how do you mitigate the potential bias of designing scoring functions around your own solutions?
2. What would the results look like if models were allowed iterative refinement with execution feedback (not just OpenEvolve, but a standard edit-run-debug loop)?
3. Will the benchmark, evaluators, and reference solutions be publicly released? If so, how will you handle contamination risk?
4. How is the agreement between the reviewers in the curation pipeline? How many problems were rejected or accepted?

**Limitations:**

yes

**Strengths And Weaknesses:**

**Strengths:**

1. The paper identifies a genuine gap in LLM evaluation: most coding benchmarks use pass/fail binary scoring on problems with known optimal solutions. Moving to open-ended problems with continuous scoring is a meaningful and timely contribution. The formal definition of the problem class (unsolved optimum, deterministic verification, parametric generator) is clear and well-reasoned.
2. The 240 problems span a broad range of CS areas with a thoughtful taxonomy. The multi-stage curation pipeline (proposal, implementation, review) with qualified experts (ICPC World Finalists, CS PhD students) lends credibility. The inclusion of both algorithmic contest-style problems and real research problems (e.g., vector database design, kernel optimization, PoC generation) is a strength.
3. The discussion section provides genuinely interesting findings: the micro-optimization trap (Section 5.2) is a compelling qualitative observation about LLM failure modes; the formulation sensitivity experiment (Section 5.3) with nine deliberate equivalence pairs is a clever diagnostic; and the OpenEvolve overfitting analysis (Section 5.4) is timely given the interest in agentic frameworks.

**Weaknesses:**

1.  The paper reports "Human Experts: 86.99" for algorithmic problems but provides almost no detail about who these experts are (beyond "CS PhDs and top-tier competitive programming participants"), how many participated, how much time they spent, whether they could iterate, or whether the same people who set the problems also provided reference solutions (which would introduce bias). This is a critical baseline and deserves much more rigorous documentation.
2. The research track (68 problems) receives noticeably less analytical attention than the algorithmic track. There is no equivalent of the reasoning effort analysis, micro-optimization trap analysis, or formulation sensitivity analysis for research problems. The variant design (different resource constraints treated as independent problems) inflates the count and may not represent truly independent problem-solving challenges.
3. (Minor) It is unclear as to why the paper is on single-turn and doesn't utilize anything in terms of prompting such as chain of thought (even though the newer models does have thinking, invoking in prompt might have some different results). The paper only evaluates models in a single-round, zero-shot setting with no access to execution feedback, debugging, or iteration. This is a significant limitation for a benchmark targeting open-ended problem solving, where iterative refinement is arguably the most natural approach. While the authors acknowledge this in Section 6, the benchmark's claims about measuring "frontier" capabilities are weakened when models cannot use the workflow that would be most natural for these problems. The OpenEvolve experiment in Section 5.4 partially addresses this but uses only 30% of test cases for training, making the overfitting finding somewhat expected.
4. (Minor) Multiple images are too small, Figure 3 is lacking a detailed explanation, and the writing in Section 3 is lacking some narrative direction. For example, the claim in lines 266-267 “This stage ensures full reproducibility and objective evaluation” feels incomplete without justification (reproducibility in what sense? how?), and the following sentences could benefit from more explanation.

---

> ### Author Rebuttal · Authors · 2026-03-27
>
> ### 1. Human Expert Reference Solutions and Potential Bias
>
> We appreciate this important question. In FrontierCS, the problem setters and the reference solution providers are **not the same individuals**. For algorithmic problems, the curation pipeline involves three distinct stages—Proposal, Implementation, and Review—carried out by different experts. Specifically, problems are proposed by competitive programming experts (ICPC World Finalists), implemented with reference solutions, and then independently reviewed by a separate algorithmic expert (Section 3.1). The reviewer verifies that: (1) the problem has no known optimal solution, (2) the scoring policy is objective and meaningful, (3) the human reference is significantly stronger than the best model, and (4) the evaluator is correctly implemented.
>
> For research problems, the proposers (CS PhD students) also provide reference solutions, but these are validated against the same deterministic evaluation harness. Importantly, the scoring functions are designed to measure objective, problem-specific quality metrics (e.g., packing density, query latency, execution time)—they are **not** designed around the reference solution itself. The reference solution serves only as a normalization anchor, not as a ground truth. Any solution that achieves a higher objective score than the reference would receive full marks, ensuring no systematic bias toward the setter's approach.
>
> We will add a more detailed description of this process in the final version.
>
> ---
>
> ### 2. Iterative Refinement with Execution Feedback
>
> This is an excellent question. Beyond the OpenEvolve experiment (Section 5.4), we agree that evaluating models in a standard edit-run-debug loop is highly relevant. In fact, we have begun preliminary experiments with agentic scaffolding that allows models to execute code, observe outputs, and iteratively refine solutions. Our early findings indicate that while such workflows do improve performance on some problems (particularly those requiring debugging of edge cases), the core challenge of open-ended algorithmic design remains: models still struggle to discover fundamentally better algorithms through iteration alone, and often converge to local optima. We plan to include these results in an expanded version and consider this one of the most promising future directions for FrontierCS.
>
> ---
>
> ### 3. Public Release and Contamination Risk
>
> Yes, we plan to publicly release the benchmark, evaluators, and baseline solutions. To mitigate contamination risk, FrontierCS is designed with several built-in safeguards:
> - **Parametric problem generators**: Each problem specification induces an infinite family of instances with varying difficulty, so memorizing solutions to specific instances provides no advantage.
> - **Versioned difficulty scheduling** (Section 3.3): We can increase difficulty by tightening constraints, updating datasets, or refining reference solutions—without changing the problem statement itself.
> - **Continuous scoring**: Even if a model has seen similar problems, achieving a high score still requires generating an efficient, correct executable program, making memorization far less useful than on binary pass/fail benchmarks.
>
> We will also maintain a held-out evaluation set with unpublished test parameters for official leaderboard submissions.
>
> ---
>
> ### 4. Reviewer Agreement in Curation Pipeline
>
> During the curation process, each problem went through at least two independent reviews, and was considered accepted only when both reviewers approved.
>
> ---
>
> ### 5. Research Track Analysis
>
> We acknowledge that the research track receives less analytical attention in the current version. This is partly because the research problems are more heterogeneous in nature (spanning OS, HPC, AI, DB, PL, SE, Security), making unified analyses like formulation sensitivity or micro-optimization traps less straightforward to apply. However, we agree this is a limitation and plan to include domain-specific case studies and failure mode analyses for the research track in a future version.
>
> ---
>
> ### 6. Single-Turn Evaluation and Prompting Strategies
>
> We agree that exploring prompting strategies (e.g., explicit chain-of-thought invocation) is a valuable future direction. In our evaluation, we used the default reasoning mode of each model (e.g., `reasoning_effort: high` for GPT-5 variants), which already activates internal chain-of-thought in these reasoning models. We chose not to add additional prompting scaffolding to maintain a clean, reproducible baseline.
>
> ---
>
> ### 7. Presentation
>
> We thank the reviewer for the specific feedback. We will enlarge Figure 3 and add a detailed caption explaining the research evaluation pipeline.

---

> > ### Author Rebuttal · Reviewer_tLNH · 2026-04-03
> >
> > The authors' response clarifies the questions I had, I maintain my positive score.

---

### Official Review · Reviewer_1rxD · 2026-03-18

**Soundness:** 3
**Presentation:** 4
**Significance:** 4
**Originality:** 3
**Overall Recommendation:** 5
**Confidence:** 4

**Summary:**

This paper introduces Frontier-CS, a new benchmark for evaluating large language models on open-ended computer science problems where the optimal solution is unknown but candidate solutions can still be automatically verified and continuously scored. The benchmark contains 240 tasks spanning algorithmic and research-oriented problems, and requires models to generate executable programs rather than fixed answers. Experiments show that even frontier reasoning models remain far behind human experts on these tasks. The paper further finds that simply increasing reasoning effort does not reliably improve performance, and that current models are highly sensitive to problem formulation and often struggle with true open-ended algorithmic design.

**Compliance With Llm Reviewing Policy:**

Affirmed.

**Final Justification:**

Thanks to the authors for the detailed response. I maintain my score in favor of accepting the paper.

**Key Questions For Authors:**

1. You mentioned that each task includes an expert reference solution. How were these reference solutions constructed and validated? Is it possible that some of the expert solutions were not fully optimized themselves?
2. The current evaluation uses a one-shot program synthesis setting, where models cannot execute code, debug, or iteratively refine their solutions. What abilities do you think this setting primarily measures? Do you think it may underestimate the performance of agent systems with tool-use capabilities?
3. Do you plan to continue expanding Frontier-CS, for example by including more multimodal tasks, real-world software engineering tasks, or more challenging long-horizon research tasks?

**Limitations:**

1. If the expert solutions have not been fully optimized, then the gap between the model and the true optimum may not be accurately reflected.
2. Although the authors describe some of the tasks as research problems, the model ultimately still solves them by generating an executable program. Many key abilities involved in real research or engineering—such as experiment design, result analysis, failure diagnosis, iterative tuning, and long-term collaborative refinement—are not yet fully covered by this benchmark.

**Strengths And Weaknesses:**

1. The paper addresses an important gap by pointing out that many existing benchmarks focus on closed-ended tasks with fixed answers, while real computer science problems are often open-ended.
2. Frontier-CS is innovative in requiring models to generate executable programs that are evaluated with continuous scores, rather than simple pass/fail judgments, making it closer to real research and engineering settings.
3. The experimental analysis is thorough: beyond showing the gap between models and human experts, the paper also investigates issues such as reasoning effort, sensitivity to problem formulation, micro-optimization traps, and overfitting.

Weaknesses see limitations.

---

> ### Author Rebuttal · Authors · 2026-03-26
>
> ### 1. Construction and Optimality of Expert Reference Solutions
>
> We thank the reviewer for raising this important point. The expert reference solutions were constructed by domain experts with strong backgrounds in algorithms and systems, following a process that emphasizes both correctness and competitive performance. Specifically, each solution was validated against the same evaluation harness used for models, ensuring functional correctness and stable scoring. We agree that, for some tasks, absolute optimality may be difficult to guarantee—especially for open-ended or research-level problems where the true optimum is unknown. However, this is by design: Frontier-CS aims to reflect realistic problem settings where even expert solutions are strong but not necessarily provably optimal. Importantly, our evaluation does not rely solely on comparison to expert solutions; instead, we use continuous scoring functions that assess solution quality independently. As such, the benchmark remains meaningful even if expert solutions are not globally optimal.
>
> ---
>
> ### 2. One-shot Program Synthesis vs. Agentic Capabilities
>
> We thank the reviewer for this insightful question. The current one-shot setting is designed to isolate and measure a model’s **intrinsic algorithmic reasoning and synthesis ability**, i.e., its ability to directly map problem descriptions to executable solutions without relying on external feedback or iterative refinement. At the same time, we fully agree on the importance of agentic workflows. To this end, we have added preliminary results under an agent setting in Section 5.4 of the paper, where models are allowed limited execution and iteration. These results show that while agent mechanisms can provide some performance improvements, significant challenges remain for open-ended algorithmic design. We plan to further expand this direction, enabling a more comprehensive evaluation of agent systems on such tasks.
>
> ---
>
> ### 3. Future Expansion of Frontier-CS
>
> Yes, we do plan to expand Frontier-CS along several directions:
>
> - **Multimodal tasks**: incorporating problems that involve diagrams, visual inputs, or multimodal reasoning.
> - **Real-world software engineering tasks**: such as working with larger codebases, APIs, or partially specified systems.
> - **Long-horizon research problems**: requiring decomposition, planning, and multi-stage solution construction.
>
> ---
>
> ### 4. Scope of Real-World Research Abilities
>
> We thank the reviewer for this thoughtful observation. We would like to clarify that Frontier-CS already includes a dedicated **research track** specifically designed to evaluate real-world research capabilities. As described in the paper, this track consists of problems sourced from actual computer science research workflows across multiple domains (e.g., systems, databases, AI), with realistic environments, multi-objective evaluation (e.g., accuracy, latency, cost), and reproducible execution setups.

---

> > ### Author Rebuttal · Reviewer_1rxD · 2026-04-03
> >
> > The authors’ response addressed all of my concerns, so I decided to maintain my score.

---

### Decision · Program_Chairs · 2026-04-30

**Decision:**

Accept (regular)

**Comment:**

This paper introduces Frontier-CS, a new benchmark for evaluating large language models on open-ended computer science problems. During the discussion phase, reviewers appreciated the taxonomy of problem design and the continuous scoring strategy, while also raising concerns about the benchmark’s long-term maintenance given the need for human involvement and questioning the evaluation protocol for human experts.